# Fucoidan carbon is stored in coastal vegetated ecosystems

Inga Hellige[1,2,3], Aman Akeerath Mundanatt[1,2], Jana C. Massing [1,2], Jan-Hendrik Hehemann[1]

[1]Marum Center for Environmental Research, University of Bremen, Bremen, 28359, Germany
[2]Max-Planck Institute for Marine Microbiology, Bremen, 28359, Germany
[3]Tjärnö Marine Laboratory, University of Gothenburg, Strömstad, 45296, Sweden

*Correspondence to*: Jan-Hendrik Hehemann (jhhehemann@marum.de)

**Abstract.** Coastal vegetated ecosystems are key-nature based solutions for climate change mitigation. Mangroves, seagrass
meadows and saltmarshes contribute to carbon sequestration not only through their photosynthetic activity but also by
anchoring sediments with their extensive root systems. By modulating flow coastal vegetation creates a low energy
environment for sediment that includes carbon to accumulate. These roots physically stabilize the sediment, prevent erosion
and enhance long-term retention of organic carbon. Hence, we hypothesized marine, algae derived organic matter may
especially accumulate in plant vegetated ecosystems. We used algal and plant glycans as carbon sequestration proxy to trace
the input and stabilization from source to sink and found those molecules in 92 sediment cores across different coastal vegetated
ecosystems from temperate to tropical regions. Specific monoclonal antibodies showed algal-derived fucoidans were present
in sediments of coastal vegetated ecosystems. Our findings suggest that the restoration of plant ecosystems that fix carbon
dioxide, protect coasts and enhance biodiversity should also be enumerated for the stored carbon from distant donors.
Conclusively, carbon sequestration is a synergistic outcome of photosynthetic contributors acting in concert across different
ecosystems.

## 1 Introduction

Coastal vegetated ecosystems, including mangroves, seagrass meadows, and saltmarshes, play a crucial role as carbon sinks,
storing estimated amounts of up to 0.7 Pg carbon year$^{-1}$ (Temmink et al., 2022). With carbon sequestration rates up to 10 times
greater than those of terrestrial forests (Mcleod et al., 2011), these ecosystems are increasingly recognized as nature-based
climate solutions. This high carbon sequestration capacity, coupled with their abilities to protect coastlines, support
biodiversity, and improve water quality, underscores their significance in carbon dioxide removal (CDR) strategies (Gattuso
et al., 2018; Hagger et al., 2022; Mengis et al., 2023).
Mangroves, seagrass and saltmarshes sequester carbon through biomass and sediment storage and by releasing dissolved
organic carbon (DOC) through their root system. Seagrass meadows were shown to be enriched in the sugar sucrose compared
to unvegetated areas (Sogin et al., 2022). In contrast, macro- and microalgae, which lack root systems, release most of their
carbon as exudates, constituting between 1% and 35% of their net primary production (Abdullah and Fredriksen, 2004; Paine

et al., 2021; Reed et al., 2015; Wada et al., 2007; Wada and Hama, 2013). Due to the challenges in tracking algae's carbon storage potential from source to sink, they are often excluded from blue carbon strategies (Krause-Jensen et al., 2018; Krause-Jensen and Duarte, 2016).

Despite these challenges, algae contribute substantially to blue carbon pools. Macro- and microalgae have been shown to supply up to 50% carbon in seagrass sediments (Kennedy et al., 2010) and up to 60% carbon of Red Sea mangrove sediments (Almahasheer et al., 2017), both measuring stable isotopic composition ($\delta^{13}C$) to identify algae as significant carbon donors (Krause-Jensen et al., 2018; Ortega et al., 2019). However, once released into the environment, tracking algae-derived carbon becomes complex. While a portion of this carbon is remineralized, 10-60% can persist under environmental conditions,

potentially forming long-term carbon sinks (Filbee-Dexter and Wernberg, 2020; Krause-Jensen and Duarte, 2016; Wada et al., 2008; Watanabe et al., 2020).

Further complicating this tracking process is the complexity of algae exudates. For instance, brown algae and diatoms secrete the complex and variable extracellular matrix polysaccharide fucoidan that might resist microbial degradation (Arnosti et al., 2012; Buck-Wiese et al., 2023; Giljan et al., 2023; Lloyd et al., 2022; Vidal-Melgosa et al., 2021). These substances can

assemble into particles (Huang et al., 2021; Vidal-Melgosa et al., 2021) that are carried by tides and currents before sinking to sediment (Salmeán et al., 2022; Vidal-Melgosa et al., 2022). Fucoidan might exist in dissolved, colloidal, and particle-associated forms (Buck-Wiese et al., 2023; Vidal-Melgosa et al., 2021), and with the characteristics of an extracellular polymer substance, it might transition between these stages (Chin, 1998). Particle-bound fucoidan can settle into sediments (Salmeán et al., 2022; Vidal-Melgosa et al., 2022) or gel-like substances can be resuspended back into the water column (e.g. Chin,

1998), influencing both its transport and its contribution to carbon cycling in coastal systems.

The extent of carbon transport and the composition of the stored organic carbon in coastal vegetated ecosystems remains understudied. Tracing carbon from source to sink is essential for understanding the potential for long-term carbon storage. Here, we hypothesized that polysaccharides, such as fucoidan can serve as tracers for algal carbon stored within coastal vegetated ecosystems, suggesting that blue carbon sequestration results from the collective carbon sequestration across

ecological communities.

## 2 Methods

**Sampling** 50cm deep sediment cores were collected with a Russian peat corer (5cm diameter) in saltmarsh, seagrass, mangroves and unvegetated areas around the German Bight, Malaysia and Columbia (**Fig. S1A-C**). At each location site along the German Bight, sampling was conducted in saltmarshes, seagrass beds, and unvegetated areas. In Columbia, seagrass beds,

unvegetated areas and mangroves were sampled. In Malaysia, sampling covered all four ecosystems: saltmarshes, seagrass beds, mangroves, and unvegetated areas. Ecosystems were sampled in close proximity of usually a few hundred meters to each other. Up to 4 points per ecosystem were sampled along a transect (**Fig. S1D**), in total 92 cores were analysed (**Table S1**). Sediment core samples primarily reflect the particulate organic matter (POM) pool, representing deposited and modified

material associated with particles, though some dissolved organic matter (DOM) may be sorbed or trapped within the matrix.

Porewater samples target the dissolved organic matter (DOM) pool, capturing the mobile fraction. Porewater samples were taken using lances or digging of holes at two of the sampled North Sea sites (Nessmersiel and Mettgrund). Samples were taken between 30 and 50 cm along 9 points per ecosystem (**Table S2**). Samples were directly filtered over pre-combusted (450°C, 4.5h) GMF and GF/F filters and 200 ml of filtered porewater samples were frozen at -20°C for polysaccharide analysis.

**Ecosystem classification** We categorized all sampling locations into four ecosystem types based on their dominant vegetation structure and appearance. While all ecosystems within a given site were subject to the same tidal regime, they were distinguished by clear differences in plant cover and elevation within the tidal frame.

Mangrove sites were defined as intertidal forested areas dominated by mangrove trees (e.g. *Rhizophora* sp.) forming continuous canopies with extensive aerial root systems. These zones were typically located at higher elevations within the tidal

frame.

Salt marsh sites were defined as intertidal grass-dominated habitats occurring at intermediate elevations within the tidal frame. Strong regional differences were observed between the North Sea and Baltic Sea locations. In the North Sea, saltmarshes exhibited a well-defined zonation pattern. The pioneer zone, located closest to the tidal water, was dominated by *Spartina anglica* and *Salicornia europaea*. The low marsh showed the most diversity, with *Puccinelia maritima* and *Atriplex*

*portulacoides* as dominant species, while the high marsh was characterized by *Elymus athericus*. The Baltic Sea saltmarshes lacked clear zonation and were primarily grazed habitats. Here, *Agrostis stolonifera*, *Juncus gerardii*, *Festuca rubra* were the most common species. Species were not identified for saltmarsh sites in Malaysia.

Seagrass sites were classified as subtidal to lower intertidal meadows of rooted marine angiosperms, forming continuous or patchy benthic vegetation. In the Baltic Sea *Zostera marina* was sampled and remained continuously submerged, whereas in

the North Sea area *Zostera noltii* occupied lower intertidal zones that were periodically exposed during low tide. In Malaysia, seagrass meadows were dominated by *Enhalus acoroides*, while in Colombia mixed stands of *Thalassia* sp., *Halodule* sp., and *Syringodium* sp. were present.

Unvegetated sites were defined as sediment surfaces lacking rooted macrophytes at the time of sampling. These areas included open tidal flats as well as areas where seagrass might have been previously present, but vegetation was absent during our

surveys. In such cases the sediments may still retain organic matter derived from past vegetation.

This classification was applied consistently across global sites to facilitate comparison of ecosystem types, while recognizing that local differences in vegetation and tidal ecosystems contribute to variability.

**Site description** Sampling was conducted across multiple biogeographical regions, encompassing temperate and tropical

coastal environments.

Six temperate sites were sampled along the German coast, three located along the North Sea and three along the Baltic Sea. The North Sea sites (Hamburger Hallig, Mettgrund and Nessmersiel) experience macrotidal conditions with a tidal range of

approximately 3 to 4 m. The sites comprise three distinct zones of salt marshes (pioneer zone: *Spartina anglica* and *Salicornia europaea*; low marsh: *Puccinelia maritima* and *Atriplex portulacoides;* high marsh: *Elymus athericus*) at intermediate elevations, seagrass meadows of *Zostera noltii* in the lower intertidal zone, and unvegetated tidal flats frequently inundated by tides. All three ecosystem types were sampled at each site. The Baltic Sea sites (Wendtorf, Heiligenhafen and Schleimünde) are characterized by lower salinity and limited tidal influence. Sampling focused on subtidal seagrass meadows of *Zostera marina*, which remained continuously submerged. Salt marsh zones occurred at higher elevations, dominated by *Agrostis stolonifera*, *Juncus gerardii*, *Festuca rubra*. Unvegetated sediments were sampled in close proximity to seagrass areas. In Wendtorf and Heiligenhafen all three ecosystems were sampled, whereas in Schleimünde only saltmarsh areas were sampled. Tropical sites were sampled in Malaysia and Columbia. Two sites were located along the coasts of Peninsular Malaysia: Setiu and Paka on the east coast, and Langkawi Island on the west coast. These tropical intertidal systems experience tidal regimes and warm, saline waters year-round. All sites contained mangrove forests, although species-level composition is not known. Seagrass meadows of *Enhalus acoroides* occurred at lower tidal elevations in Langkawi and Setiu, where nearby unvegetated tidal flats were also present. Salt-marsh vegetation, for which species-level information is likewise unavailable, was only observed and sampled at Setiu. The Colombian sampling area was located along the Caribbean coast south of Cartagena, characterized by warm tropical waters. The sites included mangrove stands dominated by *Rhizophora* sp. and *Laguncularia* sp., at higher elevations, and seagrass meadows composed of a mixed assemblage of *Thalassia* sp., *Halodule* sp., and *Syringodium* sp. in lower intertidal and shallow subtidal zones. Unvegetated sediments were sampled adjacent to the seagrass meadows. All three ecosystem types were represented across the sampled sites.

**Processing** Cores were split visually according to physiochemical layers into up to 5 parts. Each layer was homogenized by hand in a ziplock bag and stored at 4°C. Subsamples were frozen at -20°C prior to freeze drying with an Alpha 1-4 LSCbasic freeze dryer from Christ at -55°C until constant vacuum. Samples were pulverised with a ball mill (FRITSCH LLC Planeten-Mikromühle PULVERISETTE 7 premium line) at 14G for 3 min. Sediment powder was stored at room temperature in the dark until further analysis.

Porewater samples were run over AMICON filtration device with 5kDa membrane (Biomax) to separate the polysaccharide fraction over 5 kDa, washed 2 times with 300 µL MilliQ and up-concentrated by a factor of 5 in MilliQ-water. Samples were freeze dried and resuspended at a final concentration factor of 100 times in MilliQ-water.

**Carbohydrate extraction** 90 mg of dried and pulverised material was sequentially extracted using MilliQ-water and 0.3 M EDTA. Sediment was mixed with 1.8 mL of MilliQ-water, vortexed and kept in an ultrasonic water bath for 1 hour. Extracts were centrifuged at 6000G for 15 min. The resulting supernatant was transferred into a new vial and the pellets were mixed with 1.8 mL of 0.3 M EDTA, vortexed and kept in an ultrasonic water bath for 1 hour. Extracts were centrifuged at 6000G for 15 min and the supernatant was again transferred into a new vial. Extracts were frozen at -20° until further analysis.

**Phenol-sulfuric assay** The total carbohydrate content was determined based on (Dubois et al., 1951). In short, 100 µL of resuspended samples or extracts were mixed with 100 µL of 5% phenol solution and 500 µL of concentrated sulfuric acid, then incubate for 10 minutes at room temperature and afterwards for 20 min at 30°C. Absorbance at 490 nm was measured using Spectramax Id3 plate reader (Molecular Devices) against a glucose standard curve.

**Acid hydrolysis** Polysaccharides were acid hydrolysed into quantifiable monosaccharides by adding 300 µL 2M HCl to 300 µL of the porewater extracts and MilliQ-sediment extracts and the 20x diluted EDTA sediment extract in pre-combusted glass vials (450°C, 4.5 h). Glass vials were sealed and polysaccharides hydrolysed at 100°C for 24 h. The samples were transferred after hydrolysis to microtubes and dried in an acid-resistant vacuum concentrator (Martin Christ Gefriertrocknungsanlagen GmbH, Germany). Samples were resuspended in 300 µL MilliQ-water.

**Monosaccharide quantification** As previously described in two studies (Engel and Händel, 2011; Vidal-Melgosa et al., 2021), monosaccharides were quantified using anion exchange chromatography with pulsed amperometric detection (HPAEC-PAD). The sample analysis was conducted on a Dionex ICS-5000+ system, equipped with a CarboPac PA10 analytical column (2 x 250 mm) and a CarboPac PA10 guard column (2 x 50 mm). Neutral and amino sugars were separated applying an isocratic phase using 18 mM NaOH. A gradient reaching up to 200 mM NaCH3COO was applied to separate acidic monosaccharides.

**Microarray** MilliQ-water and EDTA sediment extracts were equally combined, 30 µL were transferred into wells of 384-microwell plates, where two consecutive two-fold dilutions were done in printing buffer (55.2% glycerol, 44% water, 0.8% Triton X-100). The microwell plates were centrifuged at 3500G for 10 min at 15 °C. Microarray printing and probing was performed as previously described in a recent study (Vidal-Melgosa et al., 2022). In detail, we screened for all relative antibody signals displayed in **Table S3**.

**ELISA** Polysaccharide were detected using enzyme-linked immunosorbent assay (ELISA) as described in the following studies (Cornuault et al., 2014; Vidal-Melgosa et al., 2021). In short, 100 µl of each combined sediment extract and the porewater samples were pipetted into a pre-coated 96 well plate and incubated overnight at 4°C. Signal was developed using primary antibodies (BAM1 for porewaters samples and BAM1 and JIM13 for sediment samples) in skim milk PBS solution at a concentration of 1:10, followed by antibody anti-rat in skim milk PBS solution at a concentration of 1:1000. Absorbance after development was measured at 450 nm using Spectramax Id3 plate reader (Molecular Devices).

**Statistical analysis** All statistical analysis were carried out using R4.4.1 (R core team (Anon, 2020), and Julia (Bezanson et al., 2017). For sediment samples, all measurements with a relative abundance of acidic sugars exceeding 50% were excluded from further calculations. Such elevated values typically occur in samples with very low absolute monosaccharide concentrations, where analytical uncertainty is comparatively high and may distort relative abundance patterns. Porewater

samples were not excluded based on this criterion, as their measurements are obtained more directly and are less affected by extraction efficiency or matrix effects. However, caution is still required when interpreting these data, since low concentrations can also lead to increased uncertainty.

Pairwise distances were calculated using the Bray-Curtis distance metric for non-metric multidimensional scaling, using the vegan package (Oksanen et al., 2001). As NMDS can be quite sensitive to (Fahimipour and Gross, 2020), we also applied diffusion maps (Coifman et al., 2005) as a nonlinear dimensionality reduction method for comparison using 1/Bray-Curtis distance to construct the similarity matrix and a threshold of 5. The eigenvectors corresponding to the smallest non-zero eigenvalues are the most interesting as they identify the directions of the largest variation, i.e. the main dimensions of the data manifolds.

To test for significant changes between ecosystems combining MiliQ-water and EDTA extracts we applied the following statistical methods using the car package (Pante and Simon-Bouhet, 2013). In case of no normal distribution, we performed the Kruskal-Wallis test, followed by the pairwise Wilcoxon test with α-correction according to Bonferroni to adjust for the inflation of type I error due to multiple testing. Bonferroni adjustment was performed as follows: p= 0.05/k, with k as the number of single hypotheses. Here, k= 6 was used for comparison among all 4 ecosystems and k= 3 for locations without mangroves. Therefore, α= 0.0083 or α= 0.0167 was considered statistically significant. When normality was met, we conducted an ANOVA, followed by a post-hoc Tukey test for pairwise comparisons.

# 3 Results

## 3.1 Global sediment analysis revealed similar monosaccharide abundance despite different ecosystems and locations

To investigate the mono- and polysaccharides stored in sediments across coastal vegetated ecosystems, we analysed 92 sediment cores from the North Sea, Baltic Sea, Malaysia and Columbia. 50cm deep sediment cores were taken in saltmarsh, seagrass, mangroves and unvegetated areas. Up to 4 points per ecosystem were sampled along a transect (**Fig. S1, Table S1**). The major building blocks of polysaccharides, including fucose, galactose, glucose, mannose, xylose and galacturonic acid were found to be shared between most ecosystems and depths (**Fig. S2, S3**). Applying a non-metric multidimensional scaling (NMDS) analysis to the dataset of relative abundances of monosaccharides in each sample, we found no distinct clusters among the 92 sediment cores, indicating the composition of each sample remains unchanged with location and ecosystem (**Fig. 1A**). Diffusion mapping the dataset of relative abundances, revealed a reverse relationship between fucose and glucose, where high fucose abundances occur with low glucose abundances and vice versa (**Fig. 1B-C, Fig. S4**).

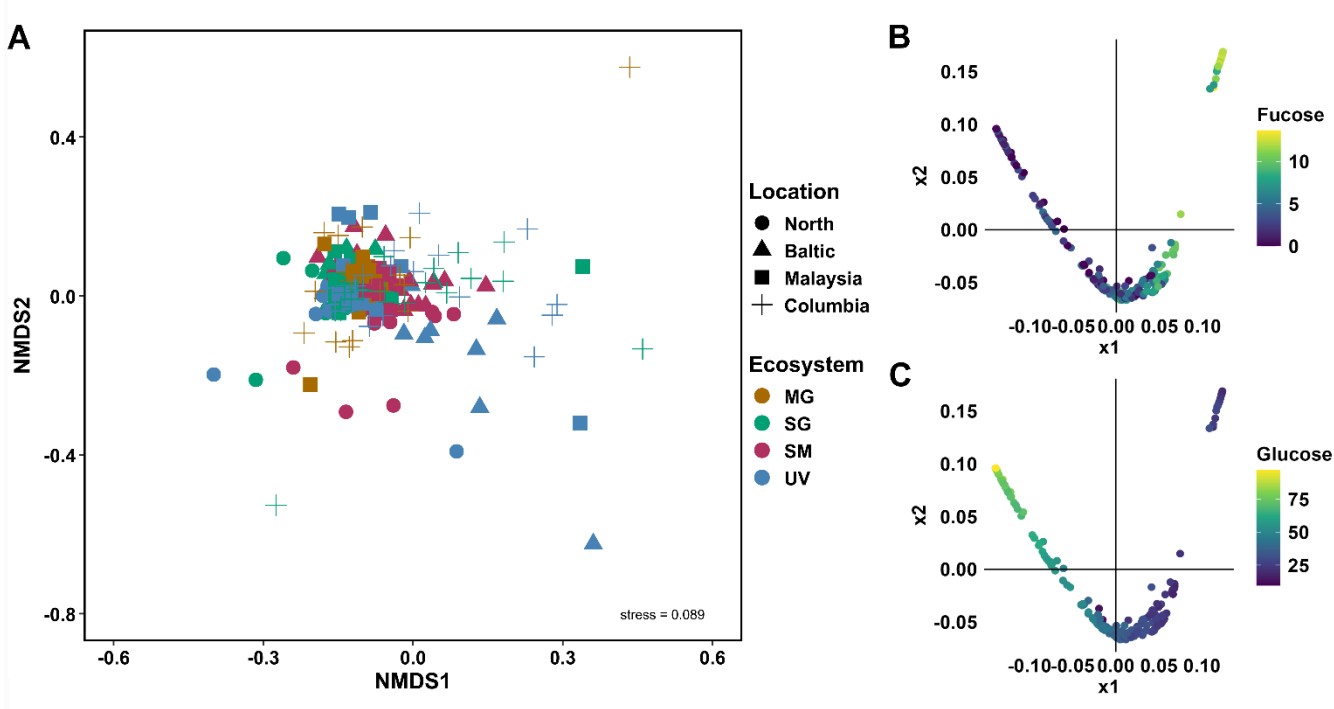

**Figure 1: 92 analysed sediment cores from temperate to tropical regions revealed no major differences in monosaccharide abundances, despite diverse ecosystems and locations. A)** Non-metric multidimensional scaling (NMDS) plot, calculated by Bray-Curtis dissimilarity of monosaccharide composition of MilliQ-water and EDTA extracts across different sediment layers for four locations, North Sea and Baltic Sea in Germany, Malaysia and Columbia for four ecosystems, mangroves (MG, brown), seagrass (SG, green), saltmarsh (SM, maroon) and unvegetated (UV, blue). NMDS stress level of 0.089. **B-C)** Diffusion maps of the relative abundances of monosaccharides identify new explanatory variables. X1 and x2 correspond to two new variables that identify the main dimensions of the data manifolds. Colouring the datapoints by relative abundance of fucose **(B)** and glucose **(C)** reveals an inverse relationship, where samples with high fucose levels tend to have low glucose levels, and vice versa.

The quantification of total carbohydrates across different ecosystems revealed the highest maximum mean concentrations of $1.67 \pm 0.23$ mg $g_{dw}^{-1}$ in saltmarsh sediments. Mangroves and saltmarshes showed significantly higher concentrations compared to seagrass and unvegetated areas ($p<0.001$) (**Fig. 2A**). Of the total carbohydrates, 10% could be attributed to total hydrolyzable carbohydrates. Except for mangroves, all ecosystems showed significant differences to each other ($p<0.001$, after Bonferroni correction, $\alpha=0.0083$) (**Fig. 2B**). Unvegetated areas ranged the lowest in mean concentrations of $35.94 \pm 1.04$ ug $g_{dw}^{-1}$, followed by seagrass with $67.65 \pm 2.32$ ug $g_{dw}^{-1}$ and saltmarsh ($250.55 \pm 9.04$ ug $g_{dw}^{-1}$) areas. Mangroves recorded the highest mean concentrations of $494.08 \pm 45.51$ ug $g_{dw}^{-1}$, but also the greatest variations (**Fig. 2B**). The quantification of total carbohydrates targets extractable and hydrolyzable carbohydrate residues and not the total amount of carbohydrate-derived carbon.

Carbohydrates incorporated into humic or proto-kerogen structures through diagenetic recombination are not detected. These concentrations typically decrease with sediment depth or age as carbohydrates are either remineralized or structurally transformed into non-hydrolyzable polymers. Thus, the higher values in mangrove and saltmarsh sediments likely reflect both higher primary production and the relative preservation of labile carbohydrate pools under these depositional conditions.

Fucose concentrations averaged $4.60 \pm 0.34$ ug $g_{dw}^{-1}$ across all samples (**Fig. 2C**). Significant differences were observed among concentrations between seagrass and unvegetated (p= 0.0066) and saltmarsh and unvegetated sediments (p<0.001, after Bonferroni correction, $\alpha = 0.0083$). Mean fucose concentrations were lowest in unvegetated areas with $2.57 \pm 0.46$ ug $g_{dw}^{-1}$, followed by seagrass with $3.88 \pm 0.40$ ug $g_{dw}^{-1}$, by mangroves with $4.35 \pm 0.67$ and highest in saltmarshes with $7.79 \pm 0.89$ ug $g_{dw}^{-1}$ (**Fig. 2C**). To reveal any monotonic relationships between sediment depths and the fucose-to-glucose ratio, we derived the Spearman rank correlation between mean depth (accounting for heterogenous depth intervals) and this ratio. This analysis revealed an increase in fucose to glucose ratio with increasing sediment depth ($R^2$= 0.4, p=0.0016; **Fig. 2D**). This trend was more evident in the rank correlation, indicating a nonlinear relationship within the dataset (**Fig. S5**).

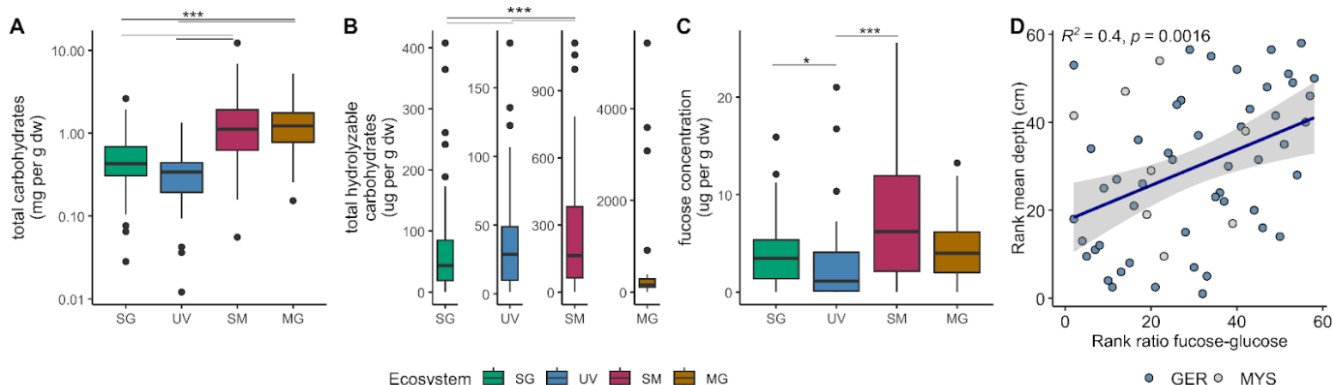

**Figure 2: Up to 10% of total hydrolyzable carbohydrates can be attributed to fucose. A)** Total carbohydrates, **B)** total hydrolyzable carbohydrates, **C)** fucose concentration in mg per g dry sediment from seagrass (SG; green), unvegetated (UV; blue), saltmarsh (SM; maroon) and mangroves (MG; yellow). **D)** Spearman rank correlation of mean depth in cm to fucose-glucose ratio (Germany: blue; Malaysia: lightgrey) indicating increase in fucose over glucose with sediment depths.

### 3.2 Specific antibody binding shows algae glycans in sediment cores of coastal vegetated ecosystems

Structure specific monoclonal antibodies analysis imply algae as a source of glycans in coastal vegetated ecosystems. Analysis of a wide range of carbohydrate microarrays (for details see **Table S3**) indicates the presence of alginate, pectin, fucoidan, arabinogalactan protein glycan, $(1\rightarrow3)$-β-D-glucan and grass-xylan (**Fig. S6**). Most signals were revealed for BAM7, indicative for alginate, fucoidan or pectin, across all ecosystems in Columbia and Germany. Due to its wide binding ability,

we tested possible correlations to monosaccharide concentrations. Mannuronic acid, being one of the main monosaccharides of alginate did not show correlation to BAM7 antibody signal (SM: $R^2 = – 0.0712$, p= 0.6618; SG: $R^2$= -0.9286, p= 0.879; not enough signals or concentrations measured for MG and UV samples; **Fig. S7A**). Guluronic acid, the other principal monosaccharide of alginate, was not measured in this study. Pectins have different monosaccharides incorporated, such as rhamnose, galacturonic acid, arabinose or galactose. Strong correlation was only observed between BAM7 signal and rhamnose concentrations in seagrass ecosystems and a slight correlation in unvegetated areas (**Fig. S7B**).

BAM7 can also cross-react with fucoidan. Testing the correlation of BAM7 signal to fucose, we found it to correlate weakly in seagrass areas and saltmarsh areas (see **Fig. S7E**). Further JIM13 indicative for arabinogalactan-protein glycans showed signal in saltmarsh and mangrove ecosystems. However, JIM13 antibody signal did not strongly correlate to the main monosaccharides of arabinogalactan, arabinose or galactose, with a weak correlation of JIM13 signal to galactose in SG ecosystems (**Fig. S8**). Notably, the upper layer of saltmarsh cores (0 up to 9.5 cm) in Heiligenhafen (HEI), Baltic Sea, revealed signal for grass xylan (LM27), not detected in further samples (**Fig. S6**). The relative antibody signal for grass xylan (LM27) shows a weak correlation to xylose concentrations ($R^2$ =0.2558, p= 0.0314; see **Fig. S9**), hinting towards its presence in some saltmarsh samples.

Although the microarray data indicated the occurrence of pectin, fucoidan and arabinogalactan-protein glycans, the signals only showed some correlation with the respective monosaccharides. The potential cross-reactivity of BAM7 and its correlation to rhamnose, hints towards the presence of pectin, rather than alginate. To verify the presence of fucoidan, arabinogalactan and pectin we applied the more sensitive ELISA method using specific antibodies (BAM1 for fucoidan, JIM13 for arabinogalactan-protein glycans and LM16 targeting rhamnogalacturonan I arabinosyl side chains, a part of pectins).

Antibody binding to arabinogalactan-protein glycan (JIM13) confirmed the signal in saltmarshes and mangroves, a lower signal in seagrass sediments and close to no signal in unvegetated areas (**Fig. S10A**). The presence of pectin was not confirmed due to high background blank signals (**Fig. S10B**). Monoclonal antibody BAM1, specific to algal polysaccharide fucoidan showed signal for all coastal vegetated ecosystems in all locations (**Fig. 3A**). Relative signals appeared to be stronger in coastal vegetated ecosystems, enriched in sediments of saltmarshes, mangroves and seagrass, slightly lower in unvegetated areas. Particle-associated fucoidan stored in sediments of all coastal ecosystems was supported by the fucoidan antibody signal positively correlating with fucose concentration (p<0.0001 for seagrass, saltmarsh and unvegetated areas; mangroves: p= 0.0022) (**Fig. 3B**).

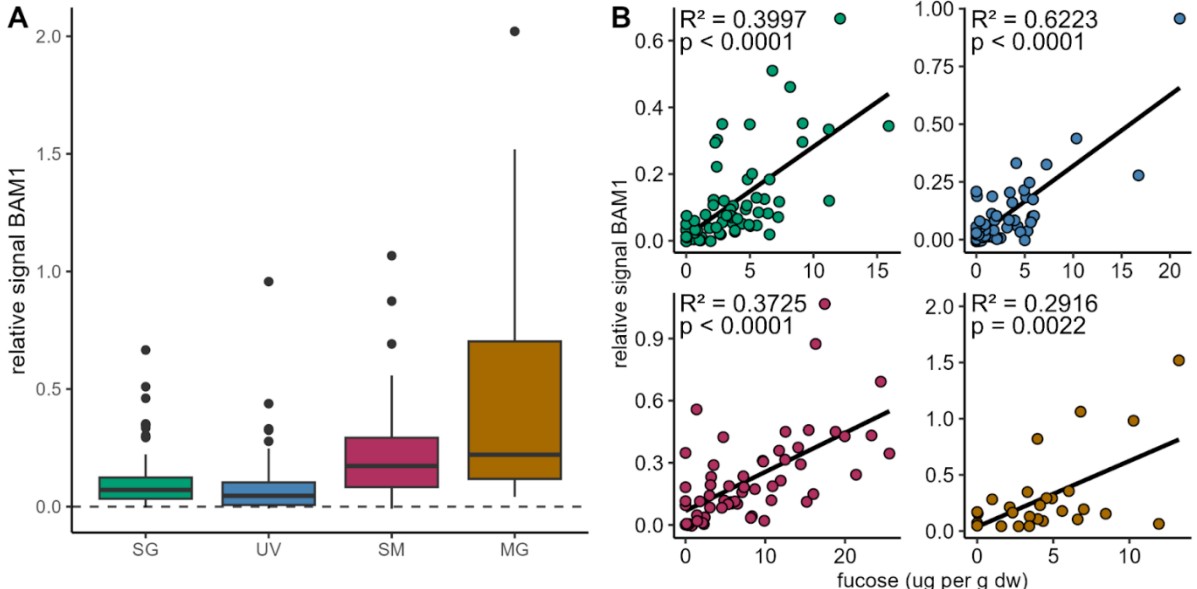

**Figure 3: Algal glycan fucoidan correlates with fucose concentrations in sediments. A)** Relative signal for antibody BAM1 (indicating presence of fucoidan) binding to sediment extracts from seagrass (SG; green), unvegetated (UV; blue), saltmarsh (SM; maroon) and mangrove areas (MG; brown). Relative signal normalized to MilliQ-water blank shown by dashed line. **B)** Relative BAM1 signal correlates with fucose concentration derived from monosaccharide analysis in ug $g_{dw}^{-1}$ (dw: dry weight).

The presence of the algal-derived glycan fucoidan was confirmed in porewater samples taken at depths of 30 to 50 cm along transects in Mettgrund (MET) and Nessmersiel (NES), North Sea sites (**Table S2**), using BAM1 antibody binding. The relative signal for fucoidan was notably higher beneath seagrass meadows compared to unvegetated areas. Similarly, strong signals were detected within the pioneer saltmarsh zone, relatively decreasing within the low saltmarsh zone and being absent inside the high saltmarsh (**Fig. 4A**). These findings align with measured fucose concentrations of $46.98 \pm 26.76$ µg $L^{-1}$ within the pioneer saltmarsh zone and $40.52 \pm 10.85$ µg $L^{-1}$ beneath seagrass meadows. Concentrations were lower in the low saltmarsh zone ($26.77 \pm 11.72$ µg $L^{-1}$) and unvegetated areas ($14.56 \pm 2.78$ µg $L^{-1}$), with no detectable fucose in the high saltmarsh zone (**Fig. 4A**). Monosaccharide abundance profiles showed again a highly similar pattern, but with distinct differences in high saltmarsh areas in Mettgrund, North Sea (**Fig. S11**).

In contrast to the porewater DOM, fucose was detected in the particulate organic matter (POM) fraction of all sediment cores, including the high saltmarsh, suggesting that at higher elevations, this material is primarily particle-associated rather than freely dissolved. This pattern supports a spatial gradient in DOM form: near the source in the pioneer and lower marsh zones, algal polysaccharides contribute to the dissolved pool, whereas in the high marsh, the same material is deposited as particles and largely retained in the sediment matrix.

285 These observations might hint towards a model in which algal-derived DOM is produced elsewhere and delivered to the marsh via tidal transport. Deposition of particle-associated material in the high marsh explains the correspondence between POM and DOM across marsh zones. These results indicate that dissolved algal-derived glycans, such as fucoidan, are secreted and transported into vegetated coastal ecosystems, which may act as acceptor systems. Algal-derived glycans carried by tidal waters may contribute to the enhancement and stabilization of sediment build-up within these ecosystems (**Fig. 4B**).

290

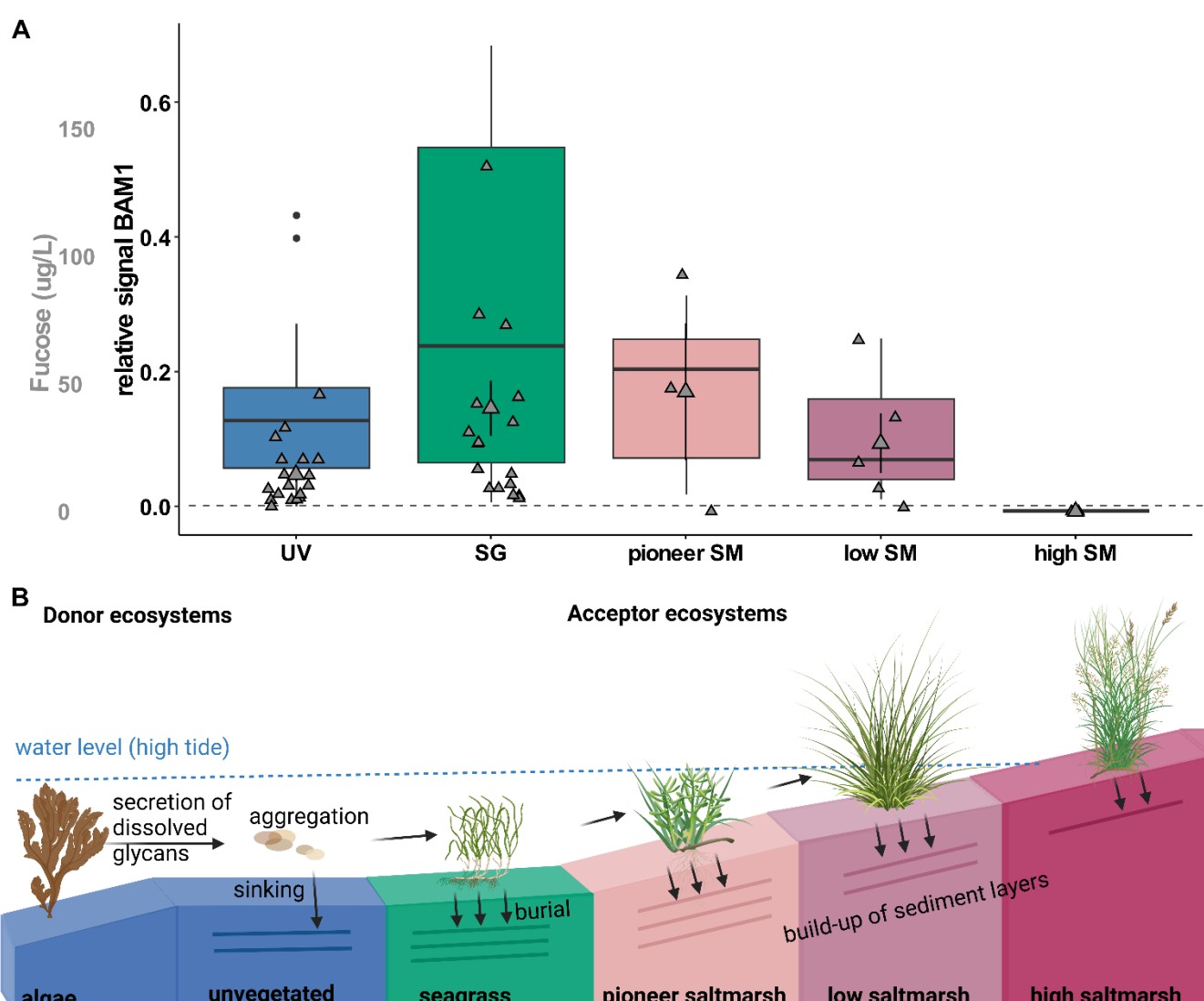

**Figure 4: Algal glycan fucoidan is stored in coastal vegetated ecosystems. A)** Relative signal for antibody BAM1 (indicating presence of fucoidan) binding to porewater extracts from unvegetated (UV; blue), seagrass (SG; green), pioneer,

low and high (SM; rose to maroon). Relative signal is normalized to signal of MilliQ-water blank (dashed line). Mean fucose concentration in $\mu g\ L^{-1}$ (mean ± s.e.m) for all ecosystems (grey). Samples taken from North Sea sites, Mettgrund (MET) and Nessmersiel (NES), see **table S2**. **B)** Schematic view on storage of algal-derived glycans in coastal vegetated ecosystems, using BioRender (Algal-Derived Carbon Transported into Coastal Vegetated Ecosystems. Created in BioRender, 2025). Donor ecosystems, such as algae secrete dissolved glycans, which may aggregate and be transported to coastal vegetated ecosystems. These acceptor ecosystems, reached by tidal waters may bury transported algal glycans and build-up of their sediment layers, accreting and storing their own and the carbon of donor ecosystems.

## 4 Discussion

By analysing the mono- and polysaccharide compositions of 92 sediment cores, we found consistent profiles across all samples, despite their origins in diverse coastal ecosystems ranging from tropical to temperate regions. This was surprising as we expected a major variation in those profiles due to the differing sources and processing pathways of DOC and POC associated with each ecosystem type. For example, up to half of mangrove net primary production is exported to the ocean as organic matter (Jennerjahn and Ittekkot, 2002) and seagrass is estimated to contribute to half of the surface sediment carbon pool of seagrass meadows (Kennedy et al., 2010). However, the observed similarity in relative monosaccharide abundances underneath all sampled ecosystems (**Fig. 1A**) suggests a glycan continuum, as previously described in surface waters (Aluwihare et al., 1997, 2002). The glycan continuum refers to the idea that structurally similar glycans persist after more labile organic matter has been broken down or organic matter has been altered to a more stable structure. Similar glycan continuum dynamics have been observed in monosaccharide patterns in the open ocean (Aluwihare et al., 1997, 2002), as well as monosaccharides derived from oligosaccharides (Bligh et al., 2024). Supported by the persistence of polysaccharides, such as arabinogalactans, mannans and sulphated fucans during a diatom bloom (Vidal-Melgosa et al., 2021) and the presence of sulphated fucans detected by antibodies in sediments (Salmeán et al., 2022; Vidal-Melgosa et al., 2022). From a carbon cycling perspective, this continuum implies that polysaccharides in deeper sediment layers may be more refractory and a partially stabilized fraction might form a long-term carbon storage. The diffusion map results highlight this consistency in glycan profiles across different ecosystems and locations (**Fig. 1B-C**). The major difference identified is driven by the relative glucose vs. fucose abundances in the samples, independent of the ecosystem and location. The fact that the relative abundance of fucose is low in high glucose samples and increases if glucose decreases indicates that fucose-rich oligosaccharides and polysaccharides might persist longer (Bligh et al., 2024; Miksch et al., 2024; Priest et al., 2023), while glucose rich glycans are broken down faster. This hypothesis is supported by the findings that glucose-rich glycans such as laminarin are much easier to break down compared to fucose-rich glycans like fucoidan, which requires a myriad of enzymes to degrade (Becker et al., 2020; Sichert et al., 2020).

Remarkably, we detected a fucoidan antibody signal in the sediments across all coastal ecosystems. As fucoidan is predominantly produced and exuded by algal species (Koch et al., 2019), it can besides other algal-derived polysaccharides serve as a tracer for algal carbon sequestration and for tracing algal carbon from sources to sinks across ecosystems. Overall,

polysaccharide screening of sediment samples revealed most binding to alginate, pectin, fucoidan and arabinogalactan (**Fig. S6, S7, Fig. 3**). Alginate, a relatively simple and linear polysaccharide that is rapidly degraded by microorganisms (Thomas et al., 2021), would not be expected to be broadly distributed or well preserved in coastal vegetated ecosystems. This is consistent with the lack of correlation between BAM7 antibody signals and one of alginates main monosaccharides mannuronic acid, suggesting instead that the BAM7 signal likely reflects the presence of pectin or fucoidan. The possible detection of pectin epitopes in sediments suggests contributions from plant biomass (Voragen et al., 2009). Arabinogalactan-protein glycan and fucoidan are exuded by algal species, in particular by brown algae (Koch et al., 2019). Arabinogalactan-protein glycans can additionally stem from plant sources (Koch et al., 2019; Vidal-Melgosa et al., 2021). Fucoidan was shown to be actively and continuously produced and secreted in the mucilage layer of brown algae and diatoms (Buck-Wiese et al., 2023; Huang et al., 2021; Vidal-Melgosa et al., 2021). Continuous fucoidan secretion increases the pool of dissolved organic matter (DOM). Particularly the contribution of gel-forming polysaccharides. may promote the formation of transparent exopolymer particles (TEP) (Engel et al., 2004). These particles enhance the aggregation of additional dissolved and particulate material, resulting in the formation of particulate organic matter (POM) (Chin, 1998).  POM can be exported through the biological carbon pump (Engel et al., 2004; Iversen, 2023). The fucoidan and arabinogalactan-protein glycan signals measured here in coastal vegetated ecosystems, suggest their export in form of DOM or POM to and storage in coastal sediments.

The consistent signal of algal-derived polysaccharides across the different ecosystems and locations and the higher abundance in coastal vegetated ecosystems reached by tidal waters compared to unvegetated areas and high saltmarsh areas (**Fig. 4**) highlights the importance of algae as donor ecosystems to blue carbon (Krause-Jensen et al., 2018; Krause-Jensen and Duarte, 2016). This finding corroborates previous evidence from stable carbon isotope ($\delta^{13}$C) studies, which demonstrated substantial inputs of externally derived microphytobenthic and other algal sources to sediment organic matter in these environments (e.g. Kennedy et al., 2010; Moncreiff and Sullivan, 2001; Volkman et al., 2008). Microphytobenthos are known to exude large quantities of extracellular carbohydrates, contributing largely to sedimentary organic matter pools (e.g. De Brouwer and Stal, 2001; Smith and Underwood, 1998). The widespread presence of algal-derived polysaccharides observed here is therefore consistent with the isotope evidence, while providing complementary molecular-level confirmation of algal carbon inputs to coastal vegetated ecosystems.

The root systems of seagrass meadows, saltmarsh plants and mangroves stabilize the sediment and the carbon stored within (e.g. Karimi et al., 2022). While around 7% of seagrass areas and up to 3% of saltmarsh and mangrove areas are lost annually (Mcleod et al., 2011), restoring and expanding these ecosystems could not only enhance local carbon storage but also increases the capacity to sequester carbon from distant algal sources, such as fucoidan.

Quantitative measurements of fucoidan-derived carbon storage will require the development of chromatographic methods capable of isolating and separating the different types of polysaccharides from sediments. Such an approach has not been achieved to the best of our knowledge yet. Future work combining fucoidan carbon quantification with $\delta^{13}$C signatures, as well as hydrodynamic measurements of the water-column fucoidan being flushed into sediments by tides has the potential to further constrain the relative contribution of fucoidan-derived carbon to sedimentary organic matter. Quantification of fucoidan carbon

in sediments is needed to provide robust estimates of this carbon pool across different locations. Our diverse sampling design enabled the detection of algal-derived carbohydrates in nearly all samples, demonstrating their widespread occurrence across coastal vegetated ecosystems.

This extensive and diverse dataset allowed us to demonstrate, for the first time, that algal-derived carbohydrates occur consistently across geographically and environmentally distinct systems. Furthermore, by including a wider panel of carbohydrate antibodies and linking these signals to their corresponding monosaccharides, we identified fucoidan as the most prevalent and traceable polysaccharide in coastal sediments. This integrative approach, combining large-scale spatial coverage with detailed biochemical analyses, provides new insights into the sources and cycling of organic matter in coastal vegetated ecosystems, thereby extending beyond the scope and conclusions of previous work.

In conclusion, coastal vegetated ecosystems store carbon dioxide partially as glycans, from which some have the potential to sequester carbon as a long-term storage. Especially the interaction of ecosystems makes it difficult to track carbon compounds back to their origin, resulting in difficulties budgeting their individual climate storage potential. Fucoidan might serve as a tracer for algal carbon stored in different locations. Overall, this study highlights the interplay and the interconnectedness of various ecosystems in storing large amounts of carbon. Together they function as a key-nature based solution for marine carbon
dioxide removal.

## 5 Acknowledgments

The authors thank technicians at Max-Planck Institute for Marine Microbiology and Marum Centre for Environmental Research, University of Bremen, namely Tina Horstmann for assistance with carbohydrate microarray analysis, Katharina Föll for HPAEC-PAD measurements, as well as Theresa Fett and Mirco Wölfelschneider from Leibniz Centre for Tropical Marine
Research (ZMT), Bremen, Germany for sampling of cores and processing of sediment cores. The authors thank Dariya Baiko and Michael Seidel from Institute for Chemistry and Biology of the Marine Environment (ICBM), Carl von Ossietzky Universität, Oldenburg and Manuel Liebeke, Jana Geuer, Bruna Imai, and Tomasz Markowski from Max-Planck Institute for Marine Microbiology for sampling and processing of porewater. The authors thank Ella Logemann and Clarisse Gösele from University of Hamburg for site specific sampling information. The authors thank José Ernesto Mancera-Pineda from
Universidad Nacional de Colombia, Esteban Zarza González from Universidad del Sinú, Colombia, Jen Nie Lee from Faculty of Science and Marine Environment, Universiti Malaysia Terengganu and A. Aldrie Amir from Institute for Environment and Development (LESTARI), Universiti Kebangsaan Malaysia. Collections along the German North Sea coast in Lower Saxony and Schleswig-Holstein were made under the permit issued by the Lower Saxony Wadden Sea National Park Authority (Ref.no.: 01.2-22249-1-1.1 (60-8) / 2022) and the National Park Administration of the State Agency for Coastal Protection,
National Park and Marine Conservation Schleswig-Holstein (Ref.no.: 3141-537.46) respectively. Collections along the German Baltic Sea coast near Massholm, Wendtorf, and Heiligenhafen were made under the permits issued by the Schleswig-Flensburg District Department for Environment Section for Nature Conservation, the Plön District Administrator, Lower

Nature Conservation Authority, Office for the Environment (Ref.no.: 3106-3/127/0161), and the District Administrator Department Nature and Soil (Ref.no.: 6.20.2-3117-IV-22-Fr) respectively. Collections from Colombia along the peninsula of

Barú were made under the permit from the Sub-Directorate for Management and Administration of Protected Areas (Ref.no.: 20222000120371). Collections from Malaysia were made under permit from the Ministry of Natural Resources, Environment and Climate Change (Access and benefit-sharing, Ref 974126). The permission to carry out research in Kilim Geoforest Park (Forest Reserve) in Kedah, Malaysia was obtained and approved by the Director-General of the Forestry Department of Peninsular Malaysia (Ref.no.: JH/ 100 Jld. 33 (75)). The permit to enter Kilim Geoforest Park (Forest Reserve) in Kedah,

Malaysia was obtained from the Langkawi District Forest Office (Permit No.: KL 72.2022). The authors acknowledge invaluable support from the Max-Planck-Society, the sea4SoCiety, CDRmare campaign in the German Marine Research Alliance.

All data is available at PANGAEA (Felden et al., 2023) data repository: https://doi.org/10.1594/PANGAEA.{975016, 977549, 979910, 979909, 979911, 979912}.

**6 Author contributions**

I.H. and J.-H.H. designed research with contributions from all authors; I.H. performed sampling in Germany and total carbohydrate, monosaccharide quantification and ELISA antibody binding. A.A.M. performed microarray analysis. I.H., J.M. and J.-H.H. analyzed data and I.H. and J.H.H. led the manuscript production with contributions from all authors.

The authors declare no competing interest.

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
