# Peer review of "Fucoidan carbon is stored in coastal vegetated ecosystems"

_EGUsphere, 2025_

## Author Response (AR1)

**Response to reviewer 1**

We thank the reviewer for their positive assessment and have worked to address each of the comments below, which we feel has greatly improved the manuscript.

Perhaps most importantly, this study integrates data from a wide diversity of global sites categorized as mangroves, salt marshes, seagrasses and unvegetated areas. Most of this data apparently showed relatively constant compositions and is lumped together into the statistical treatment on Figure 1. However, I had a hard time finding key aspects of the description of the sampling sites, especially was covered within the umbrella of "unvegetated." When I got to the Supplement, I was very surprised to learn that a single sampling site could contained all four types of system – I did not expect "salt marsh" and "mangrove" to be a few hundred meters apart from what I read in the text. This left me with a ton of questions about the elevations of transects relative to tides, plant types and coverage, spatial relationships within sites (is the mangrove upstream of the marsh? Is the unvegetated site a former seagrass bed or subaerial?) etc etc etc… and I was left unsatisfied with only the lat/long data and one "example" of a sampling site map. More precise and thorough site descriptions are needed to support the conclusions from Figs 1+2, which are highly dependent on site classification choices.

Authors response: We have added a section in the methods to clarify our chosen ecosystem classifications. Further we added a description of the different sampling sites (**lines 70-115**), where we write the following: "**Ecosystem classification** We categorized all sampling locations into four ecosystem types based on their dominant vegetation structure and appearance. While all ecosystems within a given site were subject to the same tidal regime, they were distinguished by clear differences in plant cover and elevation within the tidal frame.

[revised manuscript text omitted]

Given all of these potential axes of variability within the site, I would also hope to find a complete dataset for the raw monosaccharide abundances, to support replication / follow-up work. I didn't see these data provided, where are they? Figure 1 only shows a derived / statistical tool, so I wasn't able to actually see the data. The raw data need to be shown or at least provided in tables.

Authors response: To display the raw data of the monosaccharide abundances for all sediment cores and porewater samples, we have added supplementary figures (**Fig. S2** and **S3**).

Additionally, the complete dataset will also be made available on PANGAEA after final publication (https://doi.org/10.1594/PANGAEA.{975016, 977549, 979910, 979909, 979911, 979912).

Finally, the processes invoked to interpret the results hinge on the transfer of polysaccharides between the dissolved, weakly sorbed, and particulate pools. The paper would be strengthened by a careful clarification of which fraction of the polysaccharide pool is relevant for each data type, and an expanded discussion of these POM-DOM relationships (more below).

Authors response: We appreciate this suggestion, and we have addressed the DOM – POM relationship in more detail, please see explanations in response to the detailed comments.

**Response to detailed comments:**

Line 35 – A bit more detail in this comment would help put your new results in context. What key data types were used in the Kennedy paper (etc.) to quantify algal carbon in those environments? Is this d13C and C:N, or are there molecular tracers involved?

Authors response: Both studies used measurements of the stable isotopic composition (d13C). We have added an explanation to the section (**line 37,** please see the added text in the next response).

Because you are later going to reference different regions of the salt marsh at different tidal elevations, this would also be a useful framing for the background data presented. Do these previously published salt marsh data correspond with the upper or lower marsh?

Authors response: The previously published studies investigated seagrass meadows (Kennedy et al., 2010) and mangroves (Almahasheer et al., 2017). We addressed this comment by rewriting the text on **lines 35-37**: "Macro- and microalgae have been shown to supply up to 50% carbon in seagrass sediments (Kennedy et al., 2010) and up to 60% carbon of Red Sea mangrove sediments (Almahasheer et al., 2017), both measuring stable isotopic composition (d13C) to identify algae as significant carbon donors (Krause-Jensen et al., 2018; Ortega et al., 2019)."

Line 44 – More detail would be helpful (here or in the discussion) to lay out your working model for how fucoidan is physically transported in coastal environments. How is fucoidan (and other polysaccharides) distributed among the dissolved, sorbed, particulate (etc) phases? What do we know about how it can move in and out of the particle-associated phase?

Authors response: We appreciate this suggestion and have added the following explanation to **lines 46-50**. Here, we state the following: "Fucoidan might exist dissolved, colloidal and particle-associated forms (Buck-Wiese et al., 2023; Vidal-Melgosa et al., 2021), and with the characteristics of an extracellular polymer substance, it might transition between these stages (Chin, 1998). Particle-bound fucoidan can settle into sediments (Salmeán et al., 2022; Vidal-Melgosa et al., 2022) or gel-like substances can be resuspended back into the water column (e.g. Chin, 1998), influencing both its transport and its contribution to carbon cycling in coastal systems."

Line 127 – The axes in Figure 1B/C are derived from statistical treatments that readers may not be familiar with. Given the emergent variables x1 and x2 show this relationship between glucans

and fucans, how does it appear in the "raw" dataset? Please also show the direct relationship between the relative abundances of these compounds, at least as a supplemental figure. (Does the slope / form of this relationship between concentrations reveal anything useful?)

Authors response: The relative abundances of the two monosaccharides are derived directly from the raw dataset. The new ordering similar to NMDS identifies new variables in the dataset that have the highest variation across the dataset. The diffusion maps effectively capture nonlinear relationships within the data. We appreciate the suggestion and have added the direct relationship as a glucose-fucose plot in the supplementary material (**Fig. S4**).

Line 143 – Somewhere it would be useful to at least mention how "total carbohydrates" by your digestion method relates to carbohydrate-derived carbon in proto-kerogens, humics, and other geopolymers that have experienced recombination. Is there any trend observed in extractable carbohydrates with depth or effective age that could suggest aging affects recovery?

Authors response: We have clarified what total carbohydrates by our digestion method means in **lines 211-216:** "The quantification of total carbohydrates targets extractable and hydrolyzable carbohydrate residues and not the total amount of carbohydrate-derived carbon. Carbohydrates incorporated into humic or proto-kerogen structures through diagenetic recombination are not detected. These concentrations typically decrease with sediment depth or age as carbohydrates are either remineralized or structurally transformed into non-hydrolyzable polymers. Thus, the higher values in mangrove and saltmarsh sediments likely reflect both higher primary production and the relative preservation of labile carbohydrate pools under these depositional conditions."

We have not yet developed methods that would allow the detection of glyco-conjugates indicate of cross linking or "aging" processes. Hence such a very interesting approach should be addressed in future dedicated studies.

Line 152 – "Spearman rank correlation of mean depth" is unclear, explain exactly what is being shown? Please also show a less derived version of this relationship, i.e., between depth and the ratio itself. If the relationship is only present in the "Spearman rank" version please discuss the implications of that.

Authors response: We have clarified this point and added explanatory text (**l. 221-224**). Here we state: "To reveal any monotonic relationships between sediment depths and the fucose-to-glucose ratio, we derived the Spearman rank correlation between mean depth (accounting for heterogenous depth intervals) and this ratio. This analysis revealed an increase in fucose to glucose ratio with increasing sediment depth ($R^2$= 0.4, p=0.0016; **Fig. 2D**). This trend was more evident in the rank correlation, indicating a nonlinear relationship within the dataset (**Fig. S5**)."

In addition, following the reviewer's suggestion, we have included a supplementary figure (see **Fig. S5**) illustrating the relationship between depth and the ratio itself.

Unvegetated areas – what are these? Water depth? Tidal exposure? Nearby or upstream C sources? Grain size or sediment type? How are the unvegetated sites related to nearby vegetated sites? I imagine that a great deal of work was done to look for relationships between the data and essential these essential site variables even if strong reportable relationships weren't found.

Nonetheless it would be valuable to expand the discussion of the approach that was taken (what relationships were tested for?).

Authors response: We apologize for not explaining this more clearly in the manuscript and have added explanations in **lines 70-115**. We defined unvegetated areas as sediment surfaces lacking rooted macrophytes at the time of sampling. These areas included open tidal flats as well as areas where seagrass might have been previously present, but vegetation was absent during our surveys. In such cases the sediments may still retain organic matter derived from past vegetation. The unvegetated sites were nearby vegetated areas. Mainly we tested for differences between locations, including tidal exposure, ecosystems, species within the ecosystems or depth and sediment types.

Line 174 – The relationships in Figure 3 show EDTA extracts while Figure 4 shows porewater extracts. What do the solid extracts show (or not show) spatially, equivalent to the DOM? Is there evidence that the DOM source is actually deposited into the sediments? What is the model for its delivery to the high marsh?

Author response: Following the reviewers suggestion we highlight the following in **lines 279-288: "**In contrast to the porewater DOM, fucose was detected in the particulate organic matter (POM) fraction of all sediment cores, including the high saltmarsh, suggesting that at higher elevations, this material is primarily particle-associated rather than freely dissolved. This pattern supports a spatial gradient in DOM form: near the source in the pioneer and lower marsh zones, algal polysaccharides contribute to the dissolved pool, whereas in the high marsh, the same material is deposited as particles and largely retained in the sediment matrix.

These observations might hint towards a model in which algal-derived DOM is produced elsewhere and delivered to the marsh via tidal transport. Deposition of particle-associated material in the high marsh explains the correspondence between POM and DOM across marsh zones. These results indicate that dissolved algal-derived glycans, such as fucoidan, are secreted and transported into vegetated coastal ecosystems, which may act as acceptor systems. Algal-derived glycans carried by tidal waters may contribute to the enhancement and stabilization of sediment build-up within these ecosystems (**Fig. 4B**)."

Please also clarify which sites are included in Fig 4 in the art or its caption.

Authors response: Fig 4 shows the porewater samples taken at two locations (Mettgrund and Nessmersiel) in the North Sea area. We now clearly state this information in the caption (**l. 295-296**).

The sample type /method and its implications for which portion of the fucoidan pool is being sampled should be clarified throughout.

Authors response: We have clarified this in **lines 63-65** by stating the following: "Sediment core samples primarily reflect the particulate organic matter (POM) pool, representing deposited and modified material associated with particles, though some dissolved organic matter (DOM) may be sorbed or trapped within the matrix. Porewater samples target the dissolved organic matter (DOM) pool, capturing the mobile fraction."

Line 212 – The "glycan continuum" is an important conclusion and deserves greater explanation for your readers. What other evidence has been levied for this idea in other systems? How does it work? How could it be tested? What are its implications for coastal polysaccharides?

Authors response: We appreciate the suggestion to explain the glycan continuum in more detail. We discussed this now in more depth in lines **307-316**, where we write: "However, the observed similarity in relative monosaccharide abundances underneath all sampled ecosystems (**Fig. 1A**) suggests a glycan continuum, as previously described in surface waters (Aluwihare et al., 1997, 2002). The glycan continuum refers to the idea that structurally similar glycans persist after more labile organic matter has been broken down or organic matter has been altered to a more stable structure. Similar glycan continuum dynamics have been observed in monosaccharide patterns in the open ocean (Aluwihare et al., 1997, 2002), as well as monosaccharides derived from oligosaccharides (Bligh et al., 2024). Supported by the persistence of polysaccharides, such as arabinogalactans, mannans and sulfated fucans during a diatom bloom (Vidal-Melgosa et al., 2021) and the presence of sulphated fucans detected by antibodies in sediments (Salmeán et al., 2022; Vidal-Melgosa et al., 2022). From a carbon cycling perspective, this continuum implies that polysaccharides in deeper sediment layers may be more refractory and a partially stabilized fraction might form a long-term carbon storage."

Line 228 – Explain the proposed relationships between DOM production and POM production.

Authors response: We added the following explanation of the proposed relationships between DOM production and POM production in **lines 335-340**. "Continuous fucoidan secretion increases the pool of dissolved organic matter (DOM). Particularly the contribution of gel-forming polysaccharides. may promote the formation of transparent exopolymer particles (TEP) (Engel et al., 2004). These particles enhance the aggregation of additional dissolved and particulate material, resulting in the formation of particulate organic matter (POM) (Chin, 1998). POM can be exported through the biological carbon pump (Engel et al., 2004; Iversen, 2023). The fucoidan and arabinogalactan-protein glycan signals measured here in coastal vegetated ecosystems, suggest their export in form of DOM or POM to and storage in coastal sediments. "

Line 242 – The authors propose that fucoidan might be a tracer for exogenous polysaccharides for management purposes. So, to what extent can your dataset here give quantitative constraints on the size of this pool? Can it be compared with literature data on the net storage of C in this system? Comments on d13C implications? Is there any way to compare the accumulated flux to the source - tidal flushing volumes x reported concentrations in seawater?

Authors response: The reviewer raises very interesting points here. Unfortunately, quantification of complex polysaccharides, such as fucoidan is still challenging. Further analysis is needed for an understanding of concentrations of fucoidan carbon in sediments (see **lines 348-350 and lines 352-361**), where we write: "Quantitative measurements of fucoidan-derived carbon storage will require the development of chromatographic methods capable of isolating and separating the different types of polysaccharides from sediments. Such an approach has not been achieved to the best of our knowledge yet. […] Quantification of fucoidan carbon in sediments is needed to provide robust estimates of this carbon pool across different locations. Our diverse sampling design enabled the detection of algal-derived carbohydrates in nearly all samples, demonstrating

their widespread occurrence across coastal vegetated ecosystems. This extensive and diverse dataset allowed us to demonstrate, for the first time, that algal-derived carbohydrates occur consistently across geographically and environmentally distinct systems. Furthermore, by including a wider panel of carbohydrate antibodies and linking these signals to their corresponding monosaccharides, we identified fucoidan as the most prevalent and traceable polysaccharide in coastal sediments. This integrative approach, combining large-scale spatial coverage with detailed biochemical analyses, provides new insights into the sources and cycling of organic matter in coastal vegetated ecosystems, thereby extending beyond the scope and conclusions of previous work."

While d13C signatures have the potential to further constrain the relative contribution of fucoidan-derived carbon to sedimentary organic matter, this study did not include a paired d13C analysis of the fucoidan fraction. Future work that combines compound-specific or fraction-specific d13C measurements with fucoidan carbon quantification will help to determine variations in this polysaccharide pool. We have added this point to the discussion (**lines 350-352**). Similarly, estimating the contribution of water-column fucoidan via tidal flushing would require site-specific tidal ranges, exchange volumes, and residence times, which were beyond the scope of this study. Such calculations are feasible but would demand dedicated hydrodynamic measurements or high-resolution modelling efforts. We have therefore not attempted a precise quantification of the complete source-to-sink balance. This point has now been acknowledged and integrated into the discussion section (**lines 350-352**). Here we state: "Future work combining fucoidan carbon quantification with d13C signatures, as well as hydrodynamic measurements of the water-column fucoidan being flushed into sediments by tides has the potential to further constrain the relative contribution of fucoidan-derived carbon to sedimentary organic matter."


Authors response: Thank you for your comments regarding the novelty of our study. We disagree with the assessment that this work does not advance the field. While we used similar analytical methods to those described in (Vidal-Melgosa et al., 2022) and (Salmeán et al., 2022), our study differs substantially in scope and location. The previous studies analyzed a small number of sediment cores (three in each case) from single locations—the Red Sea and a Swedish fjord— whereas our study encompasses 92 sediment cores spanning a broad range of coastal ecosystems across temperate and tropical regions.

This extensive and diverse dataset allowed us to demonstrate, for the first time, that algal-derived carbohydrates occur consistently across geographically and environmentally distinct systems. Furthermore, by including a wider panel of carbohydrate antibodies and linking these signals to their corresponding monosaccharides, we identified fucoidan as the most prevalent and traceable polysaccharide in coastal sediments. This integrative approach, combining large-scale spatial coverage with detailed biochemical analyses, provides new insights into the sources and cycling of organic matter in coastal vegetated ecosystems, thereby extending beyond the scope and conclusions of previous work.

We agree that we should further emphasize these aspects. In response, we have added more detailed information on the sampling locations, expanded the descriptions of the studied ecosystems, strengthened the analyses linking antibody signals to monosaccharide abundances and added a statement of novelty to the discussion. We have also carefully addressed the reviewer's specific comments below.

1. Authors have analyzed the monosaccharide composition of their 93 samples but a table giving the full inventory to these compositions is not included. This would be useful to evaluate authors' statement on the fact that "fucose, galactose, glucose, mannose, xylose and galacturonic acid were found to be shared between all ecosystems and depths". In addition to this, the picture gained by the antibodies can only be shaped by the available probes. So far probes targeting sulfated galactans for instance (derived from red algae) are missing, thus the monosaccharide content is a necessary information to be displayed.

Authors response: Following the reviewer's suggestion, we have added relative monosaccharide abundances as supplementary plots (see **Fig. S2, S3**). The full data tables will be made publicly available on PANGAEA after manuscript publication (https://doi.org/10.1594/PANGAEA.{975016, 977549, 979910, 979909, 979911, 979912).

2.  The paper later focuses on the abundance of fucoidan, seen through antibody detection (BAM1) correlating a fucose content. Is there any reason why the BAM1 detection is not apparent in Fig. S2 in contrast to other antibodies? As it seems to be apparent only in Fig. 3 and beyond.

Authors response: BAM1 gave a scattered signal on the microarrays (**supplementary file, lines 59-60**). As further antibodies hint towards the presence of fucoidan, we decided to further evaluate BAM1 using the ELISA method, which offers higher sensitivity than the microarray approach. We have added an explanation in **lines 250-254**, where we write: "Although the microarray data indicated the occurrence of pectin, fucoidan and arabinogalactan-protein glycans, the signals only showed some correlation with the respective monosaccharides. The potential cross-reactivity of BAM7 and its correlation to rhamnose hints towards the presence of pectin, rather than alginate. To verify the presence of fucoidan, arabinogalactan and pectin we applied the more sensitive ELISA method using specific antibodies (BAM1 for fucoidan, JIM13 for arabinogalactan-protein glycans and LM16 targeting rhamnogalacturonan I arabinosyl side chains, a part of pectins)."

3.  In contrast, the BAM7, JIM13 and to some extent LM27 antibodies seem to give nice signals on the arrays, Fig. S2 (as discussed in the results section) but the use of these antibodies, and their most likely corresponding monosaccharides, are not included afterwards in a more comprehensive study. For instance, how are mannuronic and guluronic acid (as a proxy for alginates) correlating BAM7 recognition? A similar question can be raised for galactose/galacturonic acid and JM13, and xylose for LM27.

Authors response: We could not detect correlation between BAM 7 antibody signal and mannuronic acid concentrations. Guluronic acid, present in alginate, was not measured by the HPAEC-PAD system. We do see a weak correlation of BAM7 antibody signal to fucose concentrations. BAM7 strongly correlates to rhamnose in seagrass areas, which might hint towards presence of pectins (please see more details in the next addressed comment). JIM13 antibody signal did not correlate with galactose or arabinose and only showed a slight correlation to galactose in seagrass areas.

Further we did find a weak correlation between xylose and LM27 antibody signal, hinting towards grass xylan. We have addressed this in **lines 247-249**, where we write: "The relative antibody signal for grass xylan (LM27) shows a weak correlation to xylose concentrations ($R^2$ =0.2558, p= 0.0314; see **Fig. S9**), hinting towards its presence in some saltmarsh samples. "

4.  The results indicate a possible alginate detection across all ecosystems. Alginates being a rather simple and linear polysaccharide, easily degraded by microorganisms, one would not expect this polysaccharide to be that abundant relative to other polysaccharides. Can authors comment on this? Or could it be a cross-detection of pectins, as somehow suggested? In any case, a comparison with the uronic acid content (galacturonic, mannuronic and/or guluronic acids) would be appreciated.

Authors response: The reviewer is correct that BAM7 antibody signal might hint towards the presence of alginate. To verify this we added correlations to different monosaccharide concentrations (see **Fig. S7**). We have not measured guluronic acid with our HPAEC-PAD method. Further, we could not find any correlation to galacturonic or mannuronic acid. We have found BAM7 to positively correlate with rhamnose in seagrass samples and slightly in unvegetated areas. This suggests the presence of pectins rather than alginates. Also we see a slight correlation to fucose, which might indicate presence of fucoidan. Together with the monosaccharide correlation analysis, our results do not provide evidence for a widespread presence of alginates in coastal vegetated ecosystems. Instead, they suggest that pectins or fucoidans are more likely present. We now point this out more clearly in **lines 235-243**. Here we state the following: "Most signals were revealed for BAM7, indicative for alginate, fucoidan or pectin, across all ecosystems in Columbia and Germany. Due to its wide binding ability, we tested possible correlations to monosaccharide concentrations. Mannuronic acid, being one of the main monosaccharides of alginate did not show correlation to BAM7 antibody signal (SM: $R^2 = -0.0712$, p= 0.6618; SG: $R^2 = -0.9286$, p= 0.879; not enough signals or concentrations measured for MG and UV samples; **Fig. S7A**). Pectins have different monosaccharides incorporated, such as rhamnose, galacturonic acid, arabinose or galactose. Strong correlation was only observed between BAM7 signal and rhamnose concentrations in seagrass ecosystems and a slight correlation in unvegetated areas (**Fig. S7B**). BAM7 can also cross-react with fucoidan. Testing the correlation of BAM7 signal to fucose, we found it to correlate weakly in seagrass areas and saltmarsh areas (see **Fig. S7E**)."

5. Page 7 lines 182-185. Three sub-sampling are discussed for the porewater samples in the saltmarsh zone. This is not clear to me if similar sub-samplings have been obtained for the sediment's parts, and if so why they are not discussed? Overall, I feel that the manuscript would benefit, and get stronger conclusions, by having more information on the properties of the sampled areas (salt, oxygen levels, vegetation type, or any type of relevant information).

Authors response: We thank the reviewer for this suggestion. We have added the information on sub-samplings in saltmarsh zones. Subsampling of zones is only possible in North Sea sampling sites due to a strict vegetative zonation. This information was added to the methods (**lines 76-82**), where we write: "Salt marsh sites were defined as intertidal grass-dominated habitats occurring at intermediate elevations within the tidal frame. Strong regional differences were observed between the North Sea and Baltic Sea locations. In the North Sea, saltmarshes exhibited a well-defined zonation pattern. The pioneer zone, located closest to the tidal water, was dominated by *Spartina anglica* and *Salicornia europaea.* The low marsh showed the most diversity, with *Puccinelia maritima* and *Atriplex portulacoides* as dominant species, while the high marsh was characterized by *Elymus athericus*. The Baltic Sea saltmarshes lacked clear zonation and were primarily grazed habitats. Here, *Agrostis stolonifera*, *Juncus gerardii*, *Festuca rubra* were the most common species. Species were not identified for saltmarsh sites in Malaysia."

Further we now compare the porewater and sediment samples in the result section in more detail (**lines 279-283**): "In contrast to the porewater DOM, fucose was detected in the particulate organic matter (POM) fraction of all sediment cores, including the high saltmarsh, suggesting that at

higher elevations, this material is primarily particle-associated rather than freely dissolved. This pattern supports a spatial gradient in DOM form: near the source in the pioneer and lower marsh zones, algal polysaccharides contribute to the dissolved pool, whereas in the high marsh, the same material is deposited as particles and largely retained in the sediment matrix."

We added all information on vegetation type and sampling areas in **lines 70-115**, where we write: "**Ecosystem classification** We categorized all sampling locations into four ecosystem types based on their dominant vegetation structure and appearance. While all ecosystems within a given site were subject to the same tidal regime, they were distinguished by clear differences in plant cover and elevation within the tidal frame.

[revised manuscript text omitted]

**Minor comments**

Salmeán et al. 2022 indicated the high content of mixed linkage glucans in their Fjord sediments, and to some extent cellulose and xylan. Is there any reason why the antibody labelling targeting such components are not included in this study (especially knowing that glucose seems to be detected at high concentrations in some cases)? -or is the screening giving no detection?

Authors response: The screening gave no detection. For clarity, we have included a list of all tested antibodies, see **table S3**.

Page 3 lines 90-95. Please indicate proper antibody sources as some are not included in the referenced paper, Vidal-Melgosa et al. 2022. In particular for Fig. S2: what LM6-M and LM6 stand for, are they different antibodies?

Authors response: We apologize for not indicating the proper antibody sources and have added a list of all tested antibodies and their origins in **table S3**. LM6 and LM6-M both target $(1{\rightarrow}5)$-α-L-arabinan. LM6 recognizes short arabinan chains (Cornuault et al., 2014), while LM6-M recognizes larger arabinan chains (Cornuault et al., 2017).

Page 4 line 105. Could authors explain why "the relative abundance of acidic sugars of over 50 % were excluded from calculations"?

Authors response: Following the reviewer's suggestion, we have now added these explanations in **lines 163-168**, where we write: "For sediment samples, all measurements with a relative abundance of acidic sugars exceeding 50% were excluded from further calculations. Such elevated values typically occur in samples with very low absolute monosaccharide concentrations, where analytical uncertainty is comparatively high and may distort relative abundance patterns. Porewater samples were not excluded based on this criterion, as their measurements are obtained more directly and are less affected by extraction efficiency or matrix effects. However, caution is still required when interpreting these data, since low concentrations can also lead to increased uncertainty."

---

## Author Response (AR2)

**Manuscript ID egusphere-2025-4715**

Dear Editor,

Thank you for accepting our manuscript, *"Fucoidan carbon is stored in coastal vegetated ecosystems"*, pending technical corrections. We are grateful to both reviewers for their thoughtful and constructive feedback, which has significantly strengthened our work.

We have carefully addressed all your comments and those of the reviewers and provide a detailed point-by-point response below.

Sincerely,

Inga Hellige, Aman Akeerath Mundanatt, Jana C. Massing and Jan-Hendrik Hehemann

**Response to editor**

We thank the editor for their scientific assessment and have addressed their comment below.

1) microphytobenthos is an important primary producers in all intertidal systems, with or without macrophytes. They are not mentioned, yet they exudate large quantities of carbohydrates and there are many studies on this.
2) your overall conclusion that external algal material is found in these systems confirm what has been well documented by 13C data in the 90 and 2000ies.

Thank you for addressing these points. We have addressed both, by stating the following in lines 342-351: "The consistent signal of algal-derived polysaccharides across the different ecosystems and locations and the higher abundance in coastal vegetated ecosystems reached by tidal waters compared to unvegetated areas and high saltmarsh areas (**Fig. 4**) highlights the importance of algae as donor ecosystems to blue carbon (Krause-Jensen et al., 2018; Krause-Jensen and Duarte, 2016). This finding corroborates previous evidence from stable carbon isotope ($\delta^{13}$C) studies, which demonstrated substantial inputs of externally derived microphytobenthic and other algal sources to sediment organic matter in these environments (e.g. Kennedy et al., 2010; Moncreiff and Sullivan, 2001; Volkman et al., 2008). Microphytobenthos are known to exude large quantities of extracellular carbohydrates, contributing largely to sedimentary organic matter pools (e.g. De Brouwer and Stal, 2001; Smith and Underwood, 1998). The widespread presence of algal-derived polysaccharides observed here is therefore consistent with the isotope evidence, while providing complementary molecular-level confirmation of algal carbon inputs to coastal vegetated ecosystems."

**Response to reviewer 1**

We appreciate the reviewer's positive assessment and provide responses to the minor comments below.

line 38 and line 350: use greek delta and superscript for d13C

Thank you, we have changed this in lines 38 and 350.

line 120: is it possible to turn the 500 rpm into G units?

It is possible to turn the 500 rpm into G units, corresponding to 14G. We have addressed this in line 120.

line 151: 3500G for consistency

Thank you for pointing this out, we have changed it to match consistency.

line 134: RT replace with room temperature

We have changed RT to room temperature.

**Response to reviewer 2**

We thank the reviewer for their positive assessment and have addressed their minor comments below.

As a comment, while authors indicate that BAM7 detection does not correlate the mannuronic acid content, this would be useful to indicate that the guluronic acid content was not measured, so the reader knows why this specific correlation is not discussed.

Thank you for pointing this out. We have addressed this in lines 239-240, where we state: "Guluronic acid, the other principal monosaccharide of alginate, was not measured in this study."

Line 245 mind typo on 'galacatose'

Thank you, we have changed this.